# Grape Pomace as a Cardiometabolic Health-Promoting Ingredient: Activity in the Intestinal Environment

**DOI:** 10.3390/antiox12040979

**Published:** 2023-04-21

**Authors:** Diego Taladrid, Miguel Rebollo-Hernanz, Maria A. Martin-Cabrejas, M. Victoria Moreno-Arribas, Begoña Bartolomé

**Affiliations:** 1Institute of Food Science Research (CIAL, CSIC-UAM), C/Nicolás Cabrera, 9, 28049 Madrid, Spain; 2Department of Agricultural Chemistry and Food Science, Faculty of Science, C/Francisco Tomás y Valiente, 7, Universidad Autónoma de Madrid, 28049 Madrid, Spain

**Keywords:** grape pomace, (poly)phenols, dietary fiber, intestinal environment, digestive enzymes, nutrient transporters, enteroendocrine hormones, intestinal barrier integrity, inflammation, oxidative stress, gut microbiota

## Abstract

Grape pomace (GP) is a winemaking by-product particularly rich in (poly)phenols and dietary fiber, which are the main active compounds responsible for its health-promoting effects. These components and their metabolites generated at the intestinal level have been shown to play an important role in promoting health locally and systemically. This review focuses on the potential bioactivities of GP in the intestinal environment, which is the primary site of interaction for food components and their biological activities. These mechanisms include (*i*) regulation of nutrient digestion and absorption (GP has been shown to inhibit enzymes such as α-amylase and α-glucosidase, protease, and lipase, which can help to reduce blood glucose and lipid levels, and to modulate the expression of intestinal transporters, which can also help to regulate nutrient absorption); (*ii*) modulation of gut hormone levels and satiety (GP stimulates GLP-1, PYY, CCK, ghrelin, and GIP release, which can help to regulate appetite and satiety); (*iii*) reinforcement of gut morphology (including the crypt-villi structures, which can improve nutrient absorption and protect against intestinal damage); (*iv*) protection of intestinal barrier integrity (through tight junctions and paracellular transport); (*v*) modulation of inflammation and oxidative stress triggered by NF-kB and Nrf2 signaling pathways; and (*vi*) impact on gut microbiota composition and functionality (leading to increased production of SCFAs and decreased production of LPS). The overall effect of GP within the gut environment reinforces the intestinal function as the first line of defense against multiple disorders, including those impacting cardiometabolic health. Future research on GP’s health-promoting properties should consider connections between the gut and other organs, including the gut-heart axis, gut-brain axis, gut-skin axis, and oral-gut axis. Further exploration of these connections, including more human studies, will solidify GP’s role as a cardiometabolic health-promoting ingredient and contribute to the prevention and management of cardiovascular diseases.

## 1. Introduction

Grape pomace (GP) is the solid residue from the winemaking process and is composed mainly of grape skins and seeds [1]. Currently, GP is probably one of the food by-products most commonly used in the formulation of dietary supplements and fortified foods [2,3]. The potential health-promoting properties of GP are mainly attributed to its (poly)phenols and dietary fiber content, among other components. It is estimated that around 60–70% of the phenolic compounds of the grape remain in the pomace after winemaking [4], accounting for 4.8–5.4% of GP dry matter [5]. GP (poly)phenols encompass non-flavonoids such as hydroxybenzoic acids (C6-C1) (i.e., gallic, protocatechuic, and vanillic acids), hydroxycinnamic acids (C6-C3) (i.e., *p*-coumaric, caffeic, and ferulic acids) and stilbenes (C6-C2-C6) (i.e., resveratrol, piceatannol, and resveratrol dimers), and flavonoids (C6-C3-C6) such as flavanols (i.e., catechins, procyanidins and polymeric proanthocyanidins or condensed tannins), anthocyanins (i.e., derivatives of malvidin, petunidin, cyanidin, peonidin and delphinidin) and flavonols (i.e., derivatives of quercetin, myricetin, and kaempferol) [6]. Dietary fiber is the major component of GP, ranging between 29 and 58% in dry weight [7,8]. The fiber present in GP is fundamentally insoluble and is mainly made up of structural polysaccharides such as cellulose, hemicelluloses (xyloglucans, arabinans, galactans, xylans, and mannans), pectins, and lignin [7]. These polysaccharides are commonly bound to (poly)phenols and other non-digestible compounds, forming what was denominated “antioxidant dietary fiber” [9].

For its use in the food industry, fresh pomace usually undergoes a process of extraction, resulting in the concentration of both (poly)phenols and dietary fiber. Extracts are further stabilized by adding encapsulation agents such as maltodextrins [10]. For example, Table 1 reports the composition of (poly)phenols and dietary fiber of four GP-derived extracts. In these products, the (poly)phenols content is normally referred to as the free phenolics solubilized in organic solvents used for their determination. In contrast, the fiber is determined as the percentage of soluble and insoluble fractions or as alcohol-insoluble residue (AIR) that might include phenolics bound to polysaccharides. These data give an idea of the GP’s compositional variability depending on the starting grapes’ diversity (variety, state of maturity, etc.), the winemaking process, and the extraction and stabilization processes [10].

GP-derived products have been proposed to manage cardiovascular risk factors, including endothelial dysfunction, inflammation, hypertension, hyperglycemia, and obesity [12]. In this sense, some human intervention studies have evidenced reductions in blood pressure (BP) between 3 and 8% after GP supplementation in individuals with various symptoms of metabolic syndrome [13,14], which reached up to 14.5% in patients who suffered from hypertension [15,16]. However, other studies have reported almost no effects on BP after supplementation with GP (poly)phenols [17,18], which could be attributed to differences in the composition of the GP-derived products, doses, targeted population, etc. There is also a growing interest in evaluating the effect of GP on the regulation of baseline hyperglycemia, considered an early defect of type 2 diabetes and one of the primary anti-diabetic targets. Several intervention studies have suggested the ability of GP-derived products to reduce plasma glucose levels [14,18,19].

Nevertheless, in a certain way, it can be said that this activity of GP as a cardiometabolic health-promoting ingredient would begin in the gastrointestinal tract as GP components (i.e., (poly)phenols and fiber) undergo extensive catabolism, mainly by the action of the intestinal microbiota, that gives rise to low-molecular-weight bioactive compounds that can be absorbed and utilized by the body [20]. Dietary (poly)phenols are metabolized by the gut microbiota and concentrated in the gut lumen, reaching biologically relevant concentrations that allow them to exert local beneficial effects [21]. Regarding GP (poly)phenols, microbial catabolism pathways of the different flavonoid classes (anthocyanins, flavonols, flavan-3-ols, etc.) are known to share similar intermediate and end products such as benzoic, cinnamic, phenylacetic, and phenylpropanoic acids [22]. These phenolic metabolites are better absorbed than their precursors, occurring in plasma and urine as phase I and phase II-conjugated derivatives, and seem primarily responsible for the positive health activities in tissues and organs (systemic effects). Hepatic metabolism is a crucial step in the biotransformation of (poly)phenols, as it enables the formation of more bioavailable and biologically active metabolites that can exert beneficial effects on various tissues and organs [21,22]. Among other mechanisms, GP (poly)phenols and their metabolites appear to modulate the cell redox state by direct and indirect mechanisms such as inhibition of oxidant enzymes, activation of enzymatic and non-enzymatic antioxidant systems, and regulation of gene expression of antioxidants by interaction with redox signaling pathways [12]. The GP fiber fraction is fermented by colonic microbiota yielding short-chain fatty acids (SCFAs) such as acetate, butyrate, and propionate. SCFAs present cardioprotective effects, including modulating blood pressure and glucolipid metabolism, promoting post-infarction cardiac repair, anti-inflammation, and maintaining the gut barrier [23]. Notably, the two components, (poly)phenols and dietary fiber, interact in the body affecting their metabolism in the colon, which in turn may affect long-term systemic health effects [24].

Therefore, our goal in this paper was to disclose the potential bioactivities of GP -as a whole- in the intestinal environment, not only as a new target of health-promoting properties at the local level but also as the beginning for further effects at the systemic level (i.e., cardiometabolic effects). After a peer review, we have identified six main targets of potential bioactivity of GP in the gut: (*i*) nutrient digestion and absorption, (*ii*) enteroendocrine gut hormones release and satiety, (*iii*) gut morphology, (*iv*) intestinal barrier integrity, (*v*) intestinal inflammatory and oxidative status, and (*vi*) gut microbiome (Figure 1). The following sections (Section 2, Section 3, Section 4, Section 5, Section 6 and Section 7) recompile the most recent studies concerning the effects of GP intake on each of these issues, and the final section (Section 8) compiles the overall conclusions from all of them.

The search strategy for the literature review focused on the effects of GP on the intestinal environment (nutrient digestion and absorption, enteroendocrine gut hormones release and satiety, gut morphology, intestinal barrier integrity, intestinal inflammatory and oxidative status, and gut microbiome). The review included publications from 2010 onwards, sourced from online databases such as Scopus, Web of Science, Wiley, ScienceDirect, SpringerLink, and Google Scholar. In instances where information was scarce or crucial to the discussion, a small number of older references were also included. The literature was searched using keywords such as grape pomace, grape marc, winemaking by-products, grape residues, nutrient digestibility, intestinal enzyme, intestinal transporter, enteroendocrine hormone, intestinal barrier integrity, inflammation, oxidative stress and gut microbiota. In vitro and in vivo studies and articles focused solely on specific molecules were excluded from the main text. The relevance of each study was evaluated using a hierarchical approach based on the title, abstract, and full manuscript.

## 2. Effects of GP on Intestinal Nutrient Digestion and Absorption

In this section, we examine the impact of GP on intestinal nutrient digestion and absorption. Rich in dietary fiber and (poly)phenols, GP influences nutrient digestion and absorption through several mechanisms: physically impeding nutrient digestion and absorption, regulating intestinal enzymes, and modulating the expression of intestinal transporters. Gaining a deeper understanding of these mechanisms will offer valuable insights into the potential health benefits of GP and its role in modulating metabolic processes and promoting cardiovascular health.

### 2.1. Physical Impediment on Nutrient Digestion and Absorption

GP is a rich source of both soluble and insoluble dietary fibers. The insoluble fraction can increase the bulk of the stool and promote food movement through the digestive tract, reducing the time available for nutrient absorption [25]. In contrast, soluble fiber can form a gel-like substance in the gut, leading to delayed gastric emptying and slower nutrient absorption [26]. While this effect can be beneficial for regulating blood sugar levels, it may also limit the bioavailability of some nutrients. The high dietary fiber content in GP can slow down the rate of glucose absorption in the small intestine by forming a viscous gel-like matrix. This hinders the accessibility of digestive enzymes to carbohydrates and glucose diffusion, resulting in a more gradual release of glucose into the bloodstream [27,28,29,30]. This mechanism contributes to better glycemic control and reduced insulin spikes [31].

The soluble fiber in GP can also interfere with the emulsification process, which is necessary for the hydrolysis of triglycerides and subsequent fat absorption [30,32]. In addition to its effects on fat absorption, GP has been shown to lower dietary cholesterol absorption through multiple mechanisms. Specifically, GP’s soluble fiber and (poly)phenols can bind to bile salts in the intestine, reducing their reabsorption and promoting their excretion [33,34]. This leads to the liver utilizing more circulating cholesterol to synthesize new bile salts, ultimately resulting in decreased serum cholesterol levels. Moreover, (poly)phenols and dietary fibers in GP have been shown to inhibit intestinal cholesterol absorption by destabilizing cholesterol micelles, thereby reducing cholesterol solubility and availability [33,35]. These mechanisms have potential implications for reducing the risk of cardiovascular diseases and improving overall lipid profiles.

### 2.2. Regulation of Intestinal Enzymes

At the intestinal lumen, enzymes (α-amylase and α-glucosidase, protease, and lipase) are implicated in the digestion of primary macronutrients: carbohydrates, proteins, and lipids, respectively (Figure 2). Numerous studies have long shown the inhibitory capacity of (poly)phenols and dietary fiber separately against intestinal enzymes. It is widely known that (poly)phenols interact with enzymes, being these interactions through van der Waals, electrostatic forces, and hydrogen, as well as hydrophobic binding [36]. Concerning dietary fiber, enzyme-inhibitory capacity might be related to the inhibitors present on the surface of the insoluble fiber as well as the trapping capacity of the porous fiber network [32,37].

However, studies about the GP capacity -as a whole- to inhibit the activity of intestinal enzymes are quite recent; it should be noted that they are mainly based on in vitro assays using microbial and animal enzymes, and GP-derived products are tested directly, without previous intestinal digestion. A recompilation of these studies for α-amylase and α-glucosidase has just been done by Cisneros-Yupanqui et al. [38]. From the ten studies recompiled [28,31,39,40,41,42,43,44,45,46], only that of Kato-Schwartz et al. [31] evaluated in vivo inhibition of α-amylase and α-glucosidase by running starch and maltose tolerance tests in rats with or without administration of a GP extract. From the revised studies, the authors concluded that GP, especially the red varieties, can be perceived as a possible source of α-amylase and α-glucosidase inhibitors. However, further investigations are necessary to understand key factors such as the bioavailability and the physiological responses to the GP components [38].

The action of GP on purified intestinal proteases and peptidases has been scarcely evaluated [40]. However, studies with complex enzymatic sources such as snake venoms indicated that the dried GP exerted inhibitory actions on proteolytic activity in conjunction with other phospholipase, hemolytic, and thrombolytic activities [47]. In relation to lipases, direct GP extracts were found to inhibit pure lipases obtained from porcine pancreas [41] and also cholesterol esterase [33], although no studies have been carried out to confirm these results in vivo.

In addition to its other effects, inhibiting carbohydrate-hydrolyzing enzymes such as α-amylases and α-glucosidases in the digestive system can help prevent glucose absorption, thereby reducing postprandial hyperglycemia. This is particularly important as postprandial hyperglycemia is a significant component of all forms of diabetes [48]. Concurrently, intestinal lipase plays an important role in triacylglycerols absorption, which is related to body weight control and obesity [49]. Therefore, this capacity of GP to inhibit intestinal enzymes has been suggested to be behind, at least partly, its cardiometabolic potential proved in vivo [50,51].

### 2.3. Regulation of Intestinal Transporters

Intestinal transporters are essential for the absorption of nutrients, including sugars, lipids, amino acids, and vitamins, from the gut lumen into the bloodstream (Figure 2). Modifying intestinal transporter expression can have significant implications for nutrient uptake, gut health, and overall cardiovascular health. GP has been reported to influence the expression of glucose transporters, particularly sodium-dependent glucose transporter 1 (SGLT1) [52], facilitative glucose transporter 2 (GLUT2) [52,53,54], and fructose transporter (GLUT5) [54], which are responsible for sugars absorption in the small intestine [24]. Studies have demonstrated that GP (poly)phenols and polysaccharide-(poly)phenol complexes can enhance the expression of these transporters, leading to improved glucose uptake and blood glucose regulation [24,55]. GP has been shown to impact the expression of lipid transporters (cholesterol and fatty acids), such as Niemann–Pick C1-like 1 (NPC1L1) [56], fatty acid binding protein 1 (FABP1), and fatty acid translocase (CD36) [57], and bile salt transporters [56,58]. Studies suggest that GP (poly)phenols may downregulate the expression of these transporters, resulting in reduced cholesterol and bile salts absorption [59], which can be beneficial in managing lipid levels and cardiovascular health.

In brief, GP’s impact on nutrient digestion and absorption is multifaceted, mediated by its high content of dietary fiber and (poly)phenols, as summarized in Table 2. GP can physically impede nutrient absorption and regulate intestinal enzymes, as mentioned above. Furthermore, GP can modulate the expression of intestinal transporters, such as SGLT1, GLUT2, and NPC1L1, affecting the absorption of sugars and lipids. These mechanisms contribute to better glycemic control and reduced serum cholesterol levels, potentially lowering the risk of cardiovascular diseases. Gaining a deeper understanding of these mechanisms will provide valuable insights into the potential health benefits of GP and its role in modulating metabolic processes and promoting cardiovascular health.

## 3. Effects of GP on Enteroendocrine Gut Hormones Release and Satiety

Numerous animal model studies have shown that GP can modulate gut hormone levels, such as glucagon-like peptide-1 (GLP-1), peptide YY (PYY), cholecystokinin (CCK), ghrelin, and glucose-dependent insulinotropic polypeptide (GIP), which play crucial roles in regulating satiety and regulating food intake [60,61,62] (Figure 3).

GP has been shown to stimulate the release of GLP-1 in vitro and in vivo [63,64], an incretin hormone that enhances insulin secretion, inhibits glucagon release and reduces glycemia [65], and promotes satiety, potentially through the activation of specific G-protein-coupled receptors [66]. Increased GLP-1 levels may contribute to enhancing satiety and reducing food intake, thereby promoting weight management. The phenolic content in GP may contribute to this effect [67,68], associated with an increased L-cell number [68].

GP has been reported to increase PYY levels [64,69], another gut hormone that plays a role in appetite regulation and energy balance. Both dietary fiber and (poly)phenols in GP may be responsible for this effect [68]. Additionally, CCK, a hormone involved in digestion and satiety, has been shown to increase following GP consumption [64,70]. GP’s high (poly)phenol content may contribute to this effect by interacting with gut receptors and stimulating CCK release.

GP has been found to decrease ghrelin levels, an appetite-stimulating hormone [55,61]. Ghrelin stimulates gastrointestinal motility, reduces fat utilization and glucose-stimulated insulin release, increases body weight, and, notably, increases appetite [62]. The mechanisms underlying this ghrelin-reducing effect may include the influence of GP bitter-sensing (poly)phenols on ghrelin secretion [63] and the inhibitory effects of dietary fiber on ghrelin release [64].

GIP is an incretin hormone secreted by the K-cells in the proximal small intestine. GIP plays a role in stimulating insulin secretion in response to food intake, particularly carbohydrate-rich meals, and modulating lipid metabolism [71]. The potential mechanism through which GP might affect GIP secretion could be via the fermentation of its dietary fiber content by gut microbiota, producing SCFAs such as acetate, propionate, and butyrate. SCFAs have been implicated in the regulation of gut hormones, including GIP [72,73]. Furthermore, GP contains bioactive (poly)phenols, which may influence GIP release [63].

Dipeptidyl peptidase-IV (DPP-IV) is an enzyme that plays a crucial role in regulating incretin hormones, such as GLP-1 and GIP. DPP-IV rapidly degrades these incretins, reducing their insulin-stimulating and appetite-suppressing effects. Inhibition of DPP-IV activity can lead to increased levels of GLP-1 and GIP, improving glucose control and promoting satiety [65]. GP and grape seed proanthocyanidin extracts have demonstrated DPP-IV inhibitory activity in vitro and in vivo [63,74], suggesting a potential role in modulating incretin hormone levels and improving satiety and glycemic control.

In summary, although the evidence on humans is still scarce, accumulating in vitro and in vivo research indicates that GP modulates gut hormones involved in regulating food intake and satiety. GP’s phenolic and fiber content can increase GLP-1, PYY, and CCK levels while decreasing ghrelin levels, potentially promoting weight management. GP may also improve glycemic control by inhibiting DPP-IV and promoting GIP secretion. Overall, these findings suggest that GP has a potential as a dietary intervention for managing weight and improving glucose control.

**Figure 3 antioxidants-12-00979-f003:**
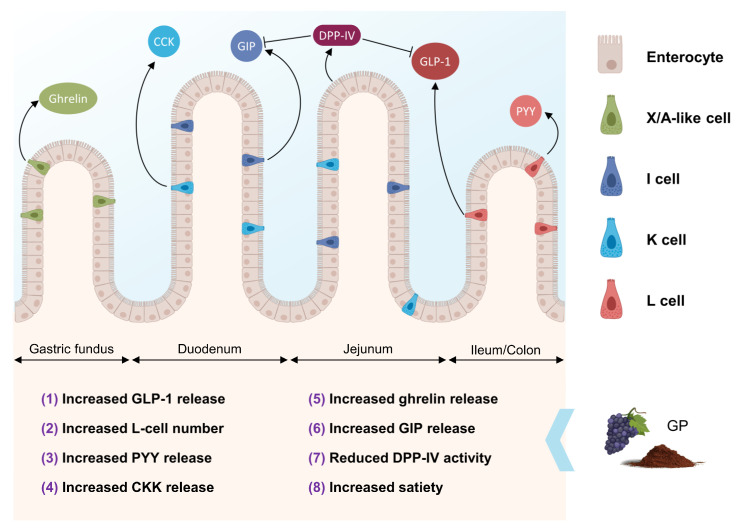
Schematic representation of the effects of GP on enteroendocrine hormone secretion. GLP-1 is secreted in L-cells in the distal ileum and colon, PYY in L-cells in the distal ileum and colon, CCK in I-cells in the duodenum and jejunum, ghrelin in X/A-like cells in the fundus of the stomach, GIP in K-cells in the duodenum and jejunum, and DPP-IV in the brush border membrane of the enterocytes primarily in the duodenum and jejunum. Numbers represent those activities associated with GP in several in vitro and in vivo experiments: **(1)**: Li et al. [63]; Casanova-Martí et al. [64]; **(2)**: Casanova-Martí et al. [68]; **(3)**: Casanova-Martí et al. [64,68]; Casanova-Martí et al. [69]; **(4)**: Casanova-Martí et al. [64]; Ginés et al. [70]; **(5)**: Serrano et al. [75]; **(6)**: Li et al. [63]; **(7)**: Li et al. [63]; González-Abuín et al. [65,74]; **(8)**: Serrano et al. [60,62,64].

## 4. Effects of GP on Gut Morphology

The gut morphology (i.e., crypt-villi structures) is a key aspect of the absorptive capacity of the intestine. The intestinal tract is lined by a single layer of epithelium originating from stem cells at the base of each crypt, giving rise to four major types of epithelial cells: absorptive enterocytes, enteroendocrine cells, which export peptide hormones, goblet cells, responsible for mucus segregation, stem cells, responsible for maintaining and repairing the tissue that lines the gut, paneth cells which secrete antimicrobial defensins, digestive enzymes, and growth factors, and tuft cells, with a role in the immune system, nutrient sensing, and gut motility regulation (Figure 4) [76]. Presumably, an impaired villus/crypt ratio implies reduced digestive capacity and imbalance in the intestinal barrier. In fact, decreased villus height/crypt depth ratio is a common response in animals treated with dextran sodium sulphate (DSS) or 2,4,6-trinitrobenzene sulfonic acid (TNBS), two inductors of intestinal damage [52,77].

In this line, several authors reported increased villus length [52,78] or villus height/crypt depth ratio [78,79,80,81] and decreased crypt depth [81,84] in the jejunum, ileum, and colon cells from rodents, lambs, pigs, and broilers fed with GP-derived products. These products were also responsible for reducing the colonic lesion and colon shortening in rats subjected to stressful DSS and TNBS treatments [77,85], with the dietary fiber and fiber-bound (poly)phenols of the GP being more effective than free (poly)phenols [85]. This response may be a consequence of the increase in villin, a calcium-regulated actin-binding protein inhabiting the intestinal brush border and related to epithelial differentiation, which in turn was responsible for the rise in goblet cell density [52]. In contrast, these outcomes enhance mucosal thickness promoted by GP products [52] and upregulation of genes involved in extracellular mucin secretion (i.e., Muc1, Muc2, and Muc3) [82,83]. Furthermore, as a secondary issue derived from the consumption of GP (poly)phenols, it has also been described the enhanced expression of genes related both to nutrient transporters in general [52] but specifically to (poly)phenols absorption [82].

The reduction in the inner surface area due to the defective development of crypt-villi structures might result in various intestinal dysfunctions and disorders such as celiac disease, inflammatory bowel disease, colorectal cancer, and brush border-related enteropathies [86]. The evidence reviewed herewith indicates that GP-derived products might help in the management of these pathologies, although, to our knowledge, there are still no specific human interventions with GP concerning them.

## 5. Effects of GP on Intestinal Barrier Integrity

As mentioned above, the intestinal barrier is a dynamic structure that separates the internal part of the host from the intestinal lumen. Of particular importance for the integrity of the intestinal barrier are the tight junctions (TJ) formed between neighboring epithelial cells, including several proteins such as occludins, claudins, junctional adhesion molecules, and plaque proteins [87] (Figure 5). The intestinal barrier selectively allows or restricts the exchange of water, ions, and macromolecules between the intestinal lumen and the underlying tissues. This exchange occurs through both transcellular pathways, governed by the cell-specific profile of transporters and channels, and paracellular pathways, mainly controlled by TJ. However, the disruption of the intestinal barrier results in what is known as the “leaky gut” and has been related to inflammation processes and intestinal dysfunction [88].

In recent years, there has been growing interest in how food-derived molecules, particularly (poly)phenols, affect the gut barrier by influencing the TJ and paracellular transport [92]. Numerous studies have investigated the effect of various grape-derived products on in vitro intestinal models such as Caco-2 cells or animal digestive tracts. These studies have demonstrated that grape-derived compounds can counteract permeabilization by studying paracellular flux (e.g., fluoroisothiocyanate-dextran (FITC-dextran)) or transepithelial electrical resistance (TEER) [93,94,95,96]. In our most recent research, we observed a decrease in paracellular transport of FITC-dextran after incubating Caco-2 monolayers with different GP-digested samples [97]. We also noted that GP-digested samples resulted in increased production of SCFAs and phenolic acids during colonic fermentation compared to control samples. This increase in SCFA and phenolic acid production may partially explain the observed decrease in permeabilization [97]. The mechanisms underlying the enhancement of the gut barrier exerted by GP are varied, including the upregulation of key genes encoding for diverse TJ proteins, such as occludins [83], claudins [83,85], ZO-1 [83,89], and TJ protein 1 [90]. Moreover, higher expression of the same proteins without variations in mRNA levels has also been reported [82,98]. Therefore, GP compounds may mediate a boost in intercellular unions through different routes than the transcription of TJ proteins, such as post-translational modifications required in TJ assembly and their association with the cytoskeleton [91] (Figure 5). However, the physiological impact of all these mechanisms is still difficult to assess.

In summary, GP-derived molecules, such as (poly)phenols, have been shown to influence the gut barrier through TJ and paracellular transport. GP (poly)phenols have a counteracting effect on permeabilization, upregulating key genes encoding for TJ proteins and mediating a boost in intercellular unions. These mechanisms result in a reduction in paracellular transport and enhanced gut barrier function.

## 6. Effects on Intestinal Inflammatory and Oxidative Status

The intestinal lumen is continuously exposed to harmful stimuli that may cause oxidative stress, inflammation, and injury. Thus, reactive oxygen species (ROS) can damage cell membranes, disturb barrier integrity, and lead to enhanced intestinal permeability, inflammation, and endotoxemia [99]. The antioxidant and anti-inflammatory properties of grape/wine (poly)phenols are widely known [100] and can be expected to be evident in the GP. Similar behavior can be said for dietary fiber, whose intake has been associated, in general, with lower systemic inflammation [101]. In this review, among the multiple and complex signaling routes involved in inflammation and oxidative stress, we have focused on those concerning the nuclear factor κB (NF-κB) for inflammation and the nuclear factor erythroid 2-related factor 2 (Nrf2) for oxidative stress since their activation cascades are considerably susceptible to GP components (Figure 6).

### 6.1. Regulation of the NF-κB Signaling Pathway

Under ordinary conditions, NF-κB is restricted to the cytoplasm forming a complex with its inhibitor (IκBα). However, several responses may initialize their decoupling through the inhibitors of kappa kinase β (IKKβ) and α (IKKα), which phosphorylate IκBα for polyubiquitination and subsequent degradation [106]. Once the NF-κB becomes free, it begins its activation and translocation to the nucleus, where it joins to specific DNA regions, the NF-κB sites [106]. This binding to DNA is responsible for expressing cytokines, adhesion molecules, and inflammatory enzymes. Among the principal stimuli that trigger the NF-κB cascade are the Toll-Like Receptors (TLR), transmembrane proteins mainly present in immune cells which react to cytokines, ROS, and, notably, to lipopolysaccharide (LPS) [107]. Overexpression of TLR is clearly recognized in various cases of intestinal bowel disease (IBD) [108]. Furthermore, through the p300 coreceptor, phosphorylated and activated by mitogen-activated protein kinase (MAPK), this enzyme participates in the inflammatory cascade since the phosphorylated NF-κB forms an active complex with the p300, which facilitates their binding to DNA [109].

Some trials feeding rodents or pigs with GP-derived products have evidenced reductions in NF-κB expression, mRNA levels, and transactivation in the duodenum [80], jejunum [52], and colon [102] cells. Moreover, Maurer et al. [103] reported downregulation of IKKβ, the kinase responsible for NF-κB liberation prior to its transport to the nucleus, and of the TLR-4 signaling, concluding that fiber-bound (poly)phenols of grape peels were mainly responsible for both responses. In accordance, Yang et al. [93] reported reduced levels of MAPK in mice colonic cells, while Pistol and collaborators observed the same pattern accompanied by the suppression of TLR2 and TLR4 genes after feeding piglets with a grape seed meal [83,102]. In addition, the last authors emphasized that not only (poly)phenols may exert an inflammatory activity, highlighting the important amount of fiber and unsaturated fatty acids in grape seed products [102]. In fact, it has been described that butyrate may suppress LPS-induced NF-κB due to G-protein coupled receptors (GPCRs) both in vitro in colonic cells and ex vivo in rodents’ large intestine [110], and specific dietary fibers present in GP voided the TLR2 cascade in human dendritic cells [111].

In addition to modulating the transcription factor NF-κB, GP may also mitigate the progression of associated inflammatory processes. Following the intracellular increase in inflammatory mediators, these molecules are released into the intestinal mucosa, initiating the breakdown of the extracellular matrix and the recruitment of innate immune cells such as neutrophils and macrophages. This cascade of events can lead to significant intestinal impairment, a pattern commonly observed in patients with IBD. In studies investigating the effects of GP products on intestinal inflammation, researchers often induce ulcerative colitis in animal models. As a primary outcome, there is a significant increase in pro-inflammatory cytokines, including interleukin (IL)-1β, IL-1α, IFN-γ, tumor necrosis factor (TNF)α, and IL-6 [89,103], adhesion molecules such as intercellular adhesion molecule 1 (ICAM1) [77], and proteins involved in the breakdown of the extracellular matrix, like those belonging to the matrix metalloproteinase (MMP) family [83]. Upon administering GP in various formulations, researchers observed a reduction in cytokine levels and a downregulation of their gene expression [77,89,103]. Additionally, anti-inflammatory cytokines such as IL-10 increased [102], and ICAM-1 and MMP9 gene expression was suppressed [77]. Neutrophil migration and infiltration into the intestinal mucosa are closely associated with myeloperoxidase (MPO) activity, a potent oxidative inducer that catalyzes the production of hypochlorous acid from Cl^−^ and H_2_O_2_ in neutrophils [112]. In line with this, some studies reported reduced colonic MPO levels in animal models of ulcerative colitis after GP consumption [78,89,103], with Maurer et al. [103] attributing this response to the fiber-bound (poly)phenols fraction. Moreover, the anti-inflammatory effects of GP have been observed in healthy animals, resulting in reduced pro-inflammatory cytokine levels and gene expression [78,102], decreased MMP2 and MMP9 activity [83], and increased levels of anti-inflammatory IL-10 and IL-4 [102,103].

### 6.2. Regulation of the Nrf2 Signaling Pathway

Oxidative stress is a physiological state that arises from an imbalance between the production of reactive oxygen species (ROS) and the body’s ability to detoxify them. ROS are highly reactive molecules that can damage lipids, proteins, and nucleic acids, leading to cellular dysfunction, tissue damage, and inflammation. Oxidative stress and inflammation are closely intertwined processes that play important roles in the development of various diseases, including gut inflammation [113]. Neutrophil infiltration into the mucosa, for example, leads to increased ROS production in the area, which in turn facilitates the migration of other immune cells to the epithelium and enhances the establishment of a niche for pro-oxidant enzymes, such as inducible nitric oxide synthase (iNOS). This enzyme catalyzes the generation of the potent oxidative peroxynitrite radical [114]. Moreover, ROS are effectors of TLRs, which can stimulate the translocation of NF-κB into the nucleus [104].

As previously mentioned, another crucial signaling pathway involves antioxidant responses mediated by Nrf2 [115]. Similarly to NF-κB, Nrf2 is sequestered in the cytoplasm by binding to a Kelch-like ECH-associated protein 1 (Keap1) [116] (Figure 6). Keap1 is an oxidation-sensitive protein that dissociates from Nrf2 when ROS levels increase in the cytoplasm, enabling Nrf2 to translocate to the nucleus and bind to the antioxidant response elements (ARE) in its target genes. This process induces the expression of antioxidant enzymes such as superoxide dismutase (SOD), catalase (CAT), or glutathione peroxidase (GPx) [117]. Apart from regulating Nrf2, Keap1 has been found to inhibit the activity of IKKβ, which is responsible for initiating the NF-κB cascade. This provides another intersection point between oxidation and inflammation. Then, the activation of Nrf2 can enhance the gut epithelium’s antioxidant capacity and reduce inflammation. Many studies examining the effects of GP on inflammation have also reported increased SOD, CAT, and GPx activity and gene expression [78,103], as well as reduced levels of ROS, nitric oxide (NO), or iNOS protein [89]. Specifically, Gessner et al. [80] conducted a study involving pigs in which they observed that the increase in antioxidant enzymes was accompanied by enhanced transactivation of both Nrf2 and NF-κB.

Considering that obesity and other associated metabolic disorders are characterized by chronic low-grade intestinal inflammation—also mediated by factors such as Nrf2 and NF-κB [118]—it is plausible that the protective effects of GP against these disorders might be achieved through the regulation of these molecular pathways at the intestinal level.

In conclusion, the antioxidant and anti-inflammatory properties of GP (poly)phenols and dietary fiber have been widely studied and shown to have potential benefits in reducing oxidative stress and inflammation in the gut. Specifically, the activation cascades of the NF-κBand the Nrf2 are susceptible to GP components, as demonstrated by studies in rodents, pigs, and humans. GP-derived products have been shown to reduce NF-κB expression and transactivation, downregulate TLR signaling, and mitigate the progression of associated inflammatory processes. Moreover, GP consumption has been associated with an increase in antioxidant enzyme activity and gene expression, as well as reduced levels of ROS, nitric oxide, or iNOS. Given the association between chronic low-grade intestinal inflammation and metabolic disorders, the regulation of these molecular pathways at the intestinal level may play a crucial role in the protective effects of GP against these disorders.

## 7. Effects of GP on the Gut Microbiome

Because of its health implications, the gut microbiome is now a thriving area of research integrating basic and clinical sciences and a priority topic worldwide [119]. Alterations in the state of the gut microbiota (dysbiosis) are increasingly linked to the incidence of non-communicable diseases [120]. Among the factors that determine our gut microbiota, diet is one of the most important and, possibly, the one we can influence the most. The dietary fiber fraction has long been recognized as a food component that most modulates gut microbiota [121]. Regarding (poly)phenols, current research suggests that they have great potential for action against microbial dysbiosis, with a favorable impact on other aspects of health related to the gut microbiome [122].

As seen in gastrointestinal digestion simulations of GP, gut microbiota extensively degrade their constituents leading to a great battery of bioaccessible phenolic metabolites (i.e., benzoic, phenylacetic, and phenylpropionic acids) as well as SCFAs mainly derived from the dietary fiber fraction [123]. In turn, these GP components and their microbial-derived metabolites can modulate the composition and functionality of the gut communities [123]. The mechanisms behind this phenolic-mediated modulation are intricate and still scarcely known, which gives rise to a considerable amount of pending work in the area. For the time being, some hypotheses point to the interference of the quorum sensing mediated by (poly)phenols, limiting the formation of pathogens biofilms [124]; others suggest that competition for nutrients favors those bacteria capable of extracting the sugars attached to (poly)phenols and fiber to obtain sugars and SCFAs, generally species associated with a probiotic effect [125].

Table 3 reports the in vivo studies reported in the literature concerning the effects of GP-derived products on gut microbiota composition. Other studies dealing with other grape products (grapes, raisins, grape seeds, wine, etc.) have not been considered in this review as they show different chemical compositions, especially concerning the fiber fraction.

The first studies reporting GP’s effect on gut microbiota aimed to assess the possible application of (poly)phenols as alternatives to antibiotics in broiler chicken production [126]. These authors found that GP concentrates seemed to be effective in increasing the ileal populations of beneficial bacteria as well as markers of healthy gut morphology (Table 3). Recently, other researchers have found similar results, further corroborating that the inclusion of GP-derived products in the diet of broiler chickens favors intestinal health without affecting their blood biochemical and immune profiles [127] (Table 3).

In rodents, Chacar et al. [128,129] specifically assessed the impact of a GP phenolic extract on gut microbiota, finding that, at the conditions used, GP supplementation could inhibit non-beneficial bacteria and potentiate the growth of probiotic ones, counteracting the adverse outcomes of aging on the gut bacterial population (Table 3). Experiments have also been conducted in mice fed high-fat diets, finding that GP supplementation appeared to ameliorate the overall metabolic profile derived from a high-fat diet [57] or to facilitate the recovery of gut microbiota after antibiotic treatment [130], both effects related to changes in some key microbial genera (Table 3).

Other studies in larger animals such as pigs [78,105] and lambs [131] corroborated that GP supplementation has no adverse effects on animal growth and can also promote the content of some beneficial bacteria in the caecum [78,105,131], as well as the enhancement of particular antioxidant [105,131], and anti-inflammatory mechanisms [78] (Table 3).

Concerning human intervention studies (Table 3), the only three studies carried out to date reported slight changes in the composition of the gut microbiota after GP supplementation in healthy women [19]; subjects exhibited at least two factors for metabolic syndrome [132,133] and subjects at high-risk cardiovascular or not [16]. No significant increase in GP-derived metabolites such as SCFAs and phenolic acids after supplementation was reported [19,132,133]. A key aspect of these studies was the high inter-individual variability observed in the body response to GP supplementation in terms of biomarkers such as blood pressure [16], fasting glucose [16,19], or insulin levels [132] (Table 3). This inter-individual variability in clinical trials with GP-derived products was related to differences in gut microbiota and miRNA [19,133]. On the other hand, it happens that the relationship between gut microbiota composition and clinical and metabolic changes derived from GP supplementation may rely more on metabolic-related bacterial communities rather than bacterial genera/family, emphasizing the ecological/functional aspects of the different communities over taxonomic aspects [16].

The differences in the extent and impact of these in vivo effects may be attributable to differences in study design (animal model, GP-derived product, doses, etc.), although it seems clear that the human gut microbiota is more stable and, therefore, more resilient to diet-induced changes. However, all these studies highlight the potential modulatory effects of GP-derived products on gut microbiota composition and open the door to studies on how GP supplementation impacts gut microbiota metabolism and functionality in relation to different pathologies. For instance, gut microbiome-modulating properties of GP seem to be behind the reduction of trimethylamine-*N*-oxide (TMAO) [134], a gut microbiota-derived metabolite recognized as strongly related to cardiovascular diseases, mainly increasing the risk of atherosclerosis development [135].

Hence, gut microbiota can degrade GP components, producing bioaccessible phenolic metabolites and SCFAs, which can modulate gut communities’ composition and functionality (Figure 7). In vivo studies have shown that GP supplementation can promote intestinal health, ameliorate metabolic profiles, enhance antioxidant and anti-inflammatory mechanisms, and increase the content of beneficial bacteria in animals. Clinical trials with GP-derived products show inter-individual variability in terms of biomarkers, emphasizing the importance of metabolic-related bacterial communities over bacterial genera/family. GP-derived products have significant potential for modulating gut microbiota composition and functionality in relation to different pathologies, including reducing the risk of non-communicable diseases such as cardiovascular disease. Further research is needed to explore the therapeutic applications of GP-derived products in promoting gut health. 

## 8. Conclusions

GP is a fascinating food by-product that has long been studied for its potential use as a dietary supplement and its protective properties against cardiovascular diseases. The health-promoting properties of GP are primarily attributed to its rich content of (poly)phenols and dietary fiber, as well as the interactions between these components. Considering that the intestinal tract is the first site of interaction for food components and their potential biological activities, this review focuses on GP’s bioactivity within the gut environment and its possible implications for cardiovascular health. Based on in vitro and in vivo studies, substantial evidence supports GP’s ability to (*i*) regulate nutrient digestion and absorption (modulating digestive enzyme action in the intestinal lumen and the expression of intestinal transporters), (*ii*) modulate gut hormones (GLP-1, PYY, CCK, ghrelin, and GIP) levels and satiety, (*iii*) reinforce gut morphology (crypt-villi structures), (*iv*) protect intestinal barrier integrity through tight junctions and paracellular transport, (*v*) modulate intestinal inflammation and oxidative stress via NF-κB and Nrf2 signaling pathways, and (*vi*) positively impact gut microbiota composition and functionality. Hence, GP promotes cardiovascular health within the intestinal environment by regulating blood lipid and glucose levels, supporting appetite regulation, reducing inflammation and oxidative stress, and fostering a beneficial gut microbiota composition. The current state of knowledge does not clearly define a primary mechanism of action for GP at the intestinal level, which may vary among individuals and pathological conditions. However, these mechanisms are evidently interconnected. For example, GP-mediated changes in microbiota composition may lead to increased production of SCFAs such as acetate, propionate, and butyrate or decreased production of bacterial LPS, which are known disruptors of gut barrier integrity and inflammation stimulants. Thus, GP’s overall effect reinforces the intestinal function as a crucial first line of defense against multiple disorders, including those impacting cardiometabolic health. In a sense, GP’s systemic bioactivity begins in the gut, which can contribute to the prevention and management of cardiovascular diseases. Future research on GP’s health-promoting properties should consider connections between the gut and other organs (gut-heart axis, gut-brain axis, gut-skin axis, oral-gut axis), further solidifying its role as a cardiometabolic health-promoting ingredient.

## Figures and Tables

**Figure 1 antioxidants-12-00979-f001:**
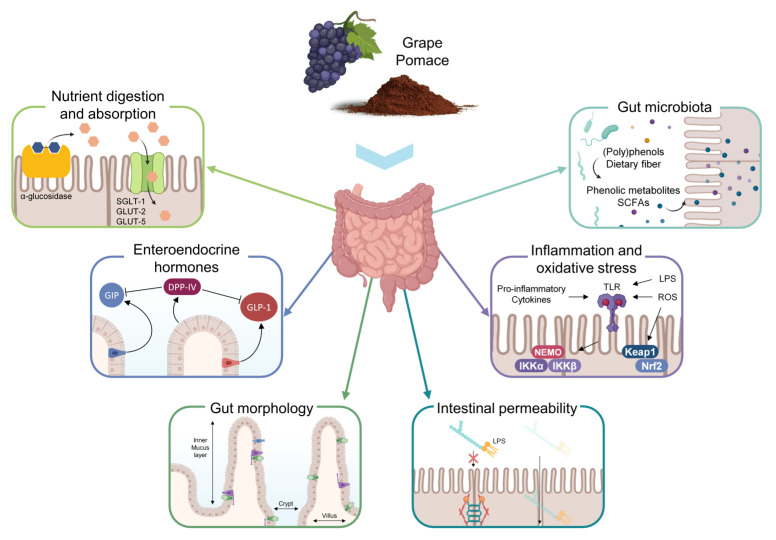
Scheme of the potential bioactivity of grape pomace (GP) in the intestinal environment.

**Figure 2 antioxidants-12-00979-f002:**
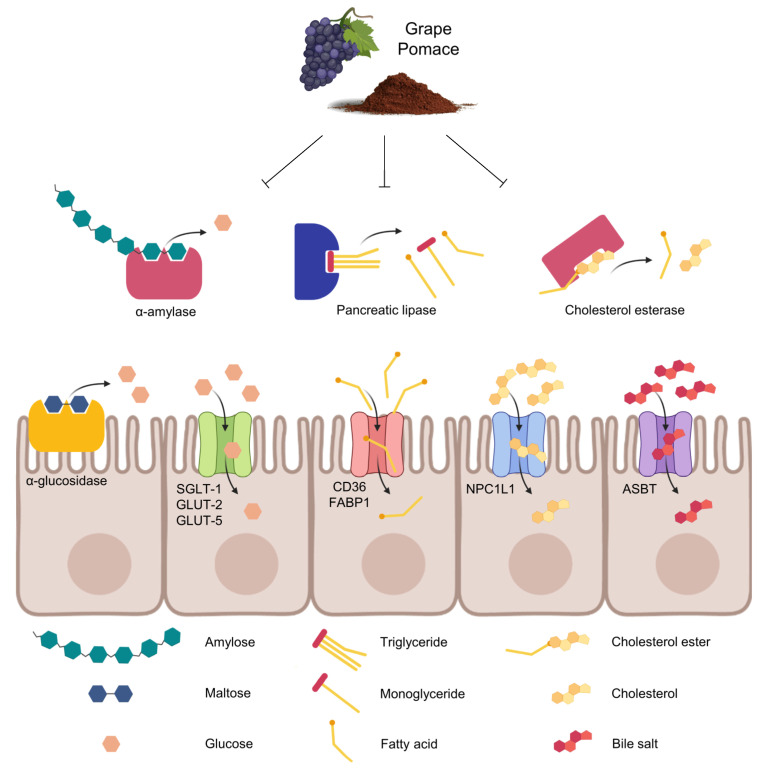
Schematic representation of the effects of GP on nutrient digestion and absorption.

**Figure 4 antioxidants-12-00979-f004:**
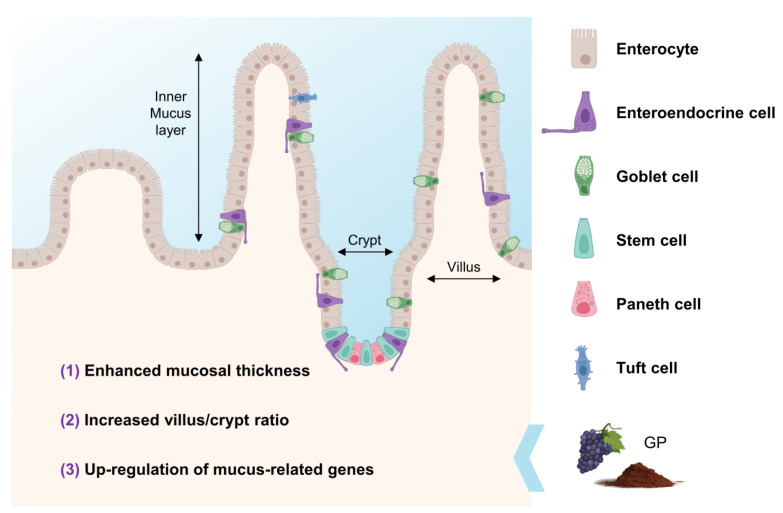
Schematic representation of villus/crypt structures in the duodenum epithelium. Numbers represent those activities linked to gut morphology associated with GP in several in vivo models: **(1)**: Bibi et al. [52]; Wang et al. [78]; **(2)**: Wang et al. [78]; Wang et al. [79]; Gessner et al. [80]; Zhang et al. [81]; **(3)**: Bibi et al. [52]; Lu et al. [82]; Pistol et al. [83].

**Figure 5 antioxidants-12-00979-f005:**
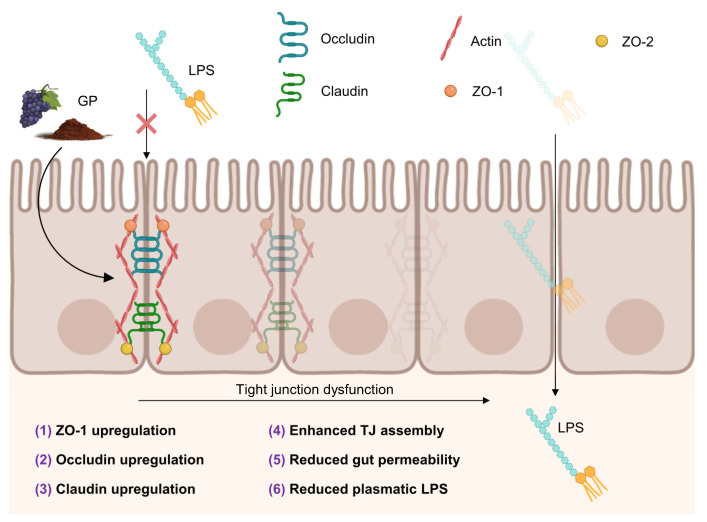
Schematic representation of intestinal epithelium (enterocytes). LPS: lipopolysaccharide; ZO: Zonula occludens or tight junction protein. Numbers represent those activities associated with GP in several in vitro and in vivo models: **(1)**: Pistol et al. [83]; Gil-Cardoso et al. [89]; **(2)**: Pistol et al. [83]; **(3)**: Maurer et al. [85]; Pistol et al. [83]; **(4)**: Nallathambi et al. [90]; Reiche and Huber [91]; **(5)**: Hidalgo-Liberona et al. [92]; Yang et al. [93]; Cremonini et al. [94]; Gil-Cardonso et al. [95]; González-Quilen et al. [96]; **(6)**: Taladrid et al. [97].

**Figure 6 antioxidants-12-00979-f006:**
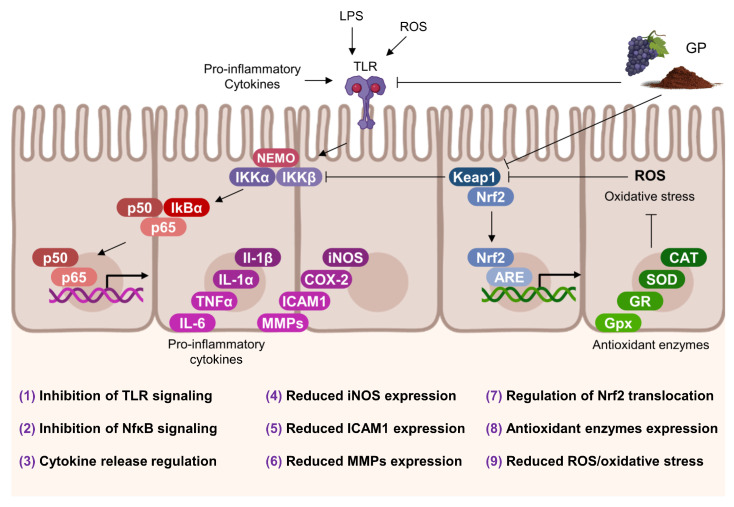
Effects of GP on intestinal inflammatory and oxidative status. A schematic representation of relevant signaling cascades mediating the gene expression of key mediators of inflammation and oxidative stress. Numbers represent those activities associated with GP in several in vitro and in vivo experiments: **(1)**: Pistol et al. [83,102]; Sheng et al. [98]; Maurer et al. [103]; **(2)**: Bibi et al. [52]; Wang et al. [104]; **(3)**: Boussenna et al. [77]; Wang at al. [78]; Gil-Cardoso et al. [89]; Nallathambi et al. [90]; Pistol et al. [102]; Maurer at al. [103]; **(4)**: Wang et al. [78]; Gil-Cardoso et al. [89] **(5)**: Boussenna et al., [77]; **(6)**: Pistol et al. [83]; **(7)**: Gessner et al. [80]; **(8)**: Wang et al. [78]; Maurer et al. [103]; **(9)**: Gessner et al. [80]; Sheng et al. [98]; Kafantaris et al. [105].

**Figure 7 antioxidants-12-00979-f007:**
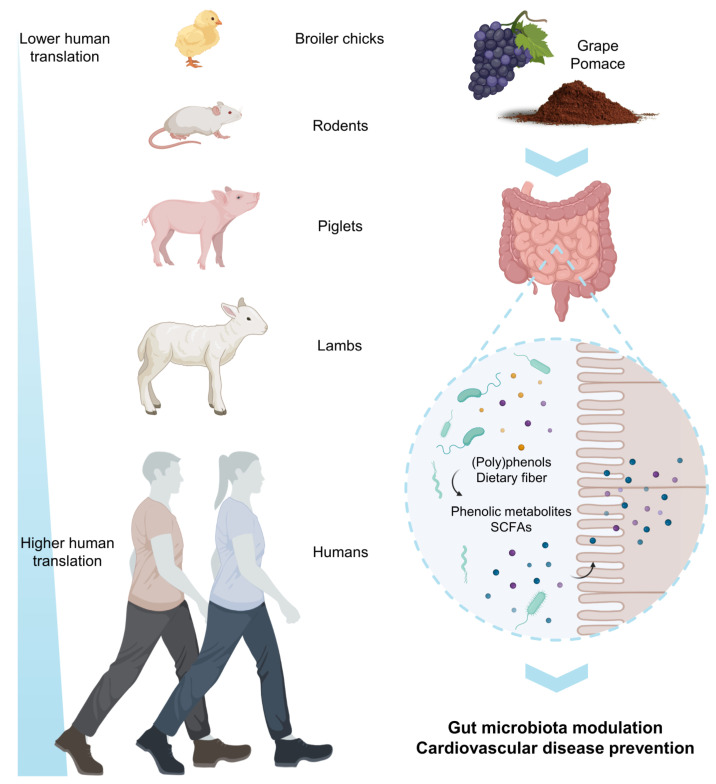
Schematic representation of the potential of GP in modulating gut microbiota composition and functionality in vivo and in clinical studies.

**Table 1 antioxidants-12-00979-t001:** Some examples of industrially manufactured GP-derived extracts: RGPE and RGPE2 (red grape pomace extracts), ORGPE (organic red grape pomace extract), and WGPE (white grape pomace extract).

	RGPE	RGPE2	ORGPE	WGPE
Total (poly)phenols (mg/g) ^a^	48.0 ± 4.1	13.0 ± 2.3	5.7 ± 0.5	3.5 ± 0.5
Phenolic acids (µg/g) ^a^	1843.0 ± 23.2	652.5 ± 12.6	853.7 ± 15.2	1211.8 ± 20.1
Flavonols (µg/g) ^a^	72.7 ± 3.5	nd	87.0 ± 7.6	nd
Flavan-3-ols (µg/g) ^a^	276.3 ± 7.6	nd	682.8 ± 8.7	295.2 ± 6.1
Anthocyanins (µg/g) ^a^	nd	nd	1706.2 ± 21.7	nd
AIR (mg/g) ^b^	930.5 ± 18.1	819.4 ± 18.9	969.8 ± 22.3	613.1 ± 21.8
Glucose (mg/g AIR) ^b, c^	446.6 ± 2.1	270.4 ± 0.6	598.4 ± 4.5	839.3 ± 1.7
Galactose (mg/g AIR) ^b, c^	28.8 ± 1.1	32 ± 0.7	36.9 ± 2.8	47.7 ± 1.1
Mannose (mg/g AIR) ^b, c^	5.9 ± 0.3	17.4 ± 0.2	4 ± 0.2	2.7 ± 0.1
Rhamnose (mg/g AIR) ^b, c^	3.2 ± 0.2	5.7 ± 0.2	2.1 ± 0.1	6.3 ± 0.1
Uronic acids (mg/g AIR) ^b, c^	62.6 ± 4.2	44.1 ± 2.9	58.1 ± 4.8	52.2 ± 2.6
Total sugars (mg/g AIR)	555.9	374.3	702.3	969.1
Klason lignin (mg/g AIR)	161 ± 9.3	22.8 ± 3.6	118.4 ± 13.3	193.6 ± 26.8

^a^ Reported in Taladrid et al. [11]. ^b^ Unpublished results. ^c^ After hydrolysis (12 M H_2_SO_4_, 3 h and room temperature, followed by 0.6 M H_2_SO_4_, 3 h and 100 °C). nd: non-detected.

**Table 2 antioxidants-12-00979-t002:** Effect of GP on nutrient digestion and absorption using in vitro and in vivo models.

Nutrient	Effect of GP on Physical Impediment	Refs.	Effect of GP on Intestinal Enzymes	Refs.	Effect of GP on Nutrient Transport	Refs.
Carbohydrates	↓ Glucose diffusion	[29]	α-amylase inhibition	[31,39,40,41,42,43,44,45]	SGLT1 downregulation	[52]
					GLUT2 downregulation	[52,53,54]
	Starch digestibility	[27,28,30]	α-glucosidase inhibition	[28,31,40,42,43,44,45,46]	GLUT5 downregulation	[54]
					↓ Intestinal glucose uptake	[24]
Lipids	↓ Triglyceride hydrolysis	[30]	Pancreatic lipase inhibition	[33,39,40,41,46]	CD36 downregulation	[57]
					FABP1 downregulation	[57]
Cholesterol	↓ Micellar cholesterol	[33]	Cholesterol esterase inhibitor	[33]	NPC1L1 downregulation	[56]
					↓ Intestinal cholesterol uptake	[59]
Bile salts	Binding	[33]	—	—	ASBT downregulation	[56,58]
Proteins	↓ Protein hydrolysis	[30]	Trypsin inhibition Chymotrypsin inhibition	[40]	PEPT1 downregulation	[54]

↓: Decrease in; SGLT1: sodium-glucose co-transporter 1; GLUT2: glucose transporter 2; GLUT5: fructose transporter; CD36: fatty acid translocase; FABP1: fatty acid-binding protein 1; NPC1L1: Niemann–Pick C1-like 1; ASBT: apical sodium-dependent bile acid transporter; PEPT1: H^+^/peptide transporter 1.

**Table 3 antioxidants-12-00979-t003:** Animal and human studies concerning the effects of GP on gut microbiota composition.

Study Pl Design/Intervention	Microbiota-Related Measurements	Changes in Microbiota Composition	Relation to Other Biomarkers	Refs.
Broiler chicks				
Cobb broiler chicks; 1-day old *n* = 25 ♂ animals/group Duration: 21 days Diets: Antibiotic-free (control) Antibiotic-included (50 mg/kg of avoparcin) Antibiotic-free diet + GP concentrate (60 g/kg)	Plate counting 16S rRNA sequencing	Ileum ↑ *Enterococcus* ↓ *Clostridium* Cecum *↑ E. coli* ↑ *Lactobacillus* ↑ *Enterococcus* ↑ *Clostridium* ↑ Bacterial diversity	=Weight gain Jejunum ↑ Villus height/crypt depth ratio	[126]
Cobb-500 broiler chicks; 1-day old *n* = 192 animals/group Duration: 42 days Diets: Antibiotic-free (control) Antibiotic-included (0.05% BMD) Antibiotic-free + 2.5% GP	16S rRNA sequencing SCFAs by GC	↑ *Bacteroidetes* ↓ *Firmicutes* ↑ *Bacteroides* ↑ *Lactobacillus* =SCFAs	↑ Feed intake =Feed conversion ratio Duodenum ↑ Villus height ↑ Villus height/crypt depth ratio Jejunum ↑ Villus height ↑ Villus width Illeum =Crypt depth =Villus height/crypt depth ratio	[127]
Rodents (rats, mice)				
Wistar rats; *n* = 6 animals/group Duration: 14 months Diets: 0.1% DMSO (control) GP phenolic extract (2.5, 5, 10, and 20 mg/kg/day)	qPCR Urine phenolic metabolites by LC-QTOF	↑ *Bifidobacterium* =*Bacteroides* =*Clostridium leptum* =*Enterococcus* ↓ *Clostridium* Cluster I		[128,129]
C57BL/6J mice; 9-week old *n* = 14 animals/group Duration: 8 weeks Diets: Control High-fat diet High-fat diet + GP phenolic extract (8.2 g/kg)	16S rRNA sequencing SCFAs by GC	↓ *Desulfovibrio* ↓ *Lactococcus* ↑ *Allobaculum* ↑ *Roseburia*	↓ Fat mass gain ↓ Adipose tissue inflammation =Food intake ↑ Glucose tolerance ↓ Insulin resistance index ↑ Antimicrobial peptides ↑ Tight junction proteins	[57]
C57BL/6J mice; 9-week old *n =* 10 ♂ animals/group Duration: 3-week antibiotics + 1 week diet Diets: Saline solution (control) GP extract 200 mg/kg	16S rRNA sequencing	Fecal microbiota ↑ Relative abundance ↑ Diversity ↑ *Akkermansia*		[130]
Piglets				
Landrace × Large White–Duroc *n* = 12 animals/group Duration: 15 days Diets: Control 4% GP	Plate counting	↑ Facultative probiotic bacteria ↑ Lactic acid bacteria ↓ *Enterobacteriacae* ↓ *Campylobacter jejuni*	↑ Weight gain ↑ GSH ↑ H_2_O_2_ decomposition activity ↑ Total antioxidant capacity ↓ MDA (TBARS) ↓ Protein carbonyls	[105]
Songliao black pigs; 28-day old *n* = 6 animals/group Duration: 28 days Diets: Control 5% GP	16S rRNA sequencing	↑ *Lactobacillus delbrueckii* ↑ *Olsenella umbonate* ↑ *Selenomonas bovis*	=Growth Jejunum ↑ Villus height ↑ Villus height/crypt depth ratio Caecum ↓ Pro-inflammatory cytokines (IL-1β, IL-8, IL-6, TNF-α) =SCFAs receptors (GPR41/43) ↑ Serum IgG	[78]
Lambs				
Chios lambs; 15-day old *n* = 12 ♂ animals/group Duration: 55 days Diets: Control 45% GP	Plate counting	↑ Facultative probiotic bacteria ↓ *Enterobacteriacae* ↓ *Escherichia coli*	↑ CAT ↑ GSH ↓ MDA (TBARS) ↓ Protein carbonyls	[131]
Humans				
Healthy women (*n* = 10) Duration: 3 weeks Diet: 1.4 g/day of a red GP extract	qPCR SCFAs by GC-MS Phenolic metabolites by UPLC-ESI-MS/MS	=Faecal bacteria populations =Faecal/urine phenolic metabolites ↑ Fecal SCFAs	↓ Blood fasting glucose Glucose metabolism-related miRNA modulation	[19]
≥2 metabolic syndrome factors subjects (*n* = 49) Duration: 6 weeks Diet: 8 g/day of GP extract	qPCR SCFAs by GC	=Microbiota profile ↑ Lactobacilliales ↑ *Bacteroides* =SCFAs	↑ insulin levels (responders)	[132]
Responders (*n* = 23) and non-responders (*n* = 26) to insulin reduction Duration: 6 weeks Diet: 8 g/day of GP extract	16S rRNA sequencing Sequencing SCFAs by GC	↓ Prevotella ↓ Firmicutes	↑ miRNA-222 levels (responders)	[133]
High-risk cardiometabolic (*n* = 17) healthy (*n* = 12) subjects Duration: 6 weeks Diets: Control 2 g/day of GP seasoning	16S rRNA sequencing SCFAs by GC-MS Phenolic metabolites by UPLC-ESI-MS/MS	=α and β diversity ↓ *Peptoniphilus* ↓ *Clostridiaceae 1* ↓ *Clostridium sensu stricto 1* ↓ *Ezakiella* ↓ *Streptococcus* ↓ *Lachnospiraceae ND3007* ↓ *Paraprevotella* ↓ *Senegalimassilia* ↑ Streptococcaceae ↓ Eggerthellaceae ↓ Coribacteriales *Incertae Sedis* ↓ Propionic acid ↑ Protocatechuic acid	↓ Blood pressure ↓ Fasting blood glucose Bacterial communities associated with cardiovascular and metabolic data	[16]

↓: Decrease in; ↑: Increase in.

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
