# Peer review of "Grape Pomace as a Cardiometabolic Health-Promoting Ingredient: Activity in the Intestinal Environment"

_antioxidants, 2023, doi:10.3390/antiox12040979_

Round 1

Reviewer 1 Report

Dear Editor and Authors,

 I am writing to inform you about my reviewer work on the paper entitled " Grape pomace as a cardiometabolic health-promoting ingredient: activity in the intestinal environment". The paper is very well written and provides critical insights into the health-promoting effects of grape pomace (GP).

As we know, GP is a winemaking by-product with numerous health benefits. The gut environment plays a crucial role in reinforcing intestinal function as the first line of defence against various disorders that affect cardiometabolic health. Therefore, exploring the bioactivities of GP in the intestinal environment is of utmost importance to understand its systemic and cardiometabolic effects.

After careful evaluation, I only found minor bibliographic errors in the paper. However, I recommend that the authors provide figures of better quality to enhance the visual impact of their work. Overall, I highly recommend the immediate publication of this paper, as it provides significant insights into the bioactivities of GP in the intestinal environment.

Based on my evaluation, I strongly recommend the immediate publication of this paper. I believe it will be of great interest to the readers of Antioxidants and make an important contribution to the field.

Thank you for considering my recommendation.

The authors have presented their research in a clear and concise manner, making it easy to follow their arguments and understand their findings.

While reading the manuscript, I noticed some minor English errors that should be corrected before publication. However, these errors do not detract from the quality of the research or the overall readability of the paper.

Author Response

Please, see attached file.

Reviewer 2 Report

The review by Taladrid et al aims to summarize the current knowledge about the health-promoting effects of grape pomace with particular emphasis on the effects at the intestinal level. The review is comprehensive of the current literature and very well-written and organized. Some indications of the positive consequences on cardiovascular diseases are indicated and explained, but I would suggest to remove the word “cardiometabolic” from the title, since the review is mostly centered on the mechanisms of action of grape pomace at intestinal level. I would also suggest to insert a short methodology section describing how the references were searched and selected for this review (instead of simply citing the peer review, which does not apply; see line 109).

Other minor comments.

Line 40. Remove “and functional foods”. Maybe better fortified foods

Line 390. Remove parenthesis

Line 564. Remove “…and its protective properties against cardiovascular diseases”, since mostly mechanisms at intestinal level have been described.

Line 569. Add possible in the following sentence “… and its possible implications for cardiovascular health”.

Author Response

Please, see attached file.

Reviewer 3 Report

General comment

The present review deals with the in vitro and in vivo biological activity of grape pomace at the intestinal level. The manuscript is substantial and is structured in a clear and understandable way in all its parts. The tables and figures accompany the reader well towards understanding the subject matter. Therefore, I recommend its publication in Antioxidants.

However, I have some suggestions to improve the quality of the review

Specific comments

Change NF-κβ into NF-κB throughout the text

Please improve the quality of the Figures, they look a bit grainy

Please report in full only the first time something is named, and subsequently only the acronym (e.g. ROS, IBD, NO etc)

Abstract

Line 13: “These components and their metabolites generated at the intestinal level”…The Authors should include hepatic phase I/II metabolites which may be predominant after GP intake.

Introduction

Line 93 – 95 please some recent references on that statement. You should also include phase I/II metabolites coming from hepatic and small intestine activity

Line 99 – 105 I would be more specific on why evaluating GP at the intestinal level, for example by highlighting how polyphenols, once taken with the diet, can concentrate precisely in the gut, reaching biologically relevant concentrations such as to be able to exert local beneficial effects

Paragraph 2.3.

In my opinion, Table 2 it should be arranged and organized differently because it is hard to read as it is.

Chapter 7

Line 517 “Other studies in larger animals such as pigs [78,105] and (?) [131]”. Please add this information

Line 525 Please remove “derived” after GP-derived

Some typos here and there have been detected, please check the manuscript

Author Response

Please, see attached file.
